# New Approaches to the Prevention of Visceral Leishmaniasis: A Review of Recent Patents of Potential Candidates for a Chimeric Protein Vaccine

**DOI:** 10.3390/vaccines12030271

**Published:** 2024-03-05

**Authors:** Diana Souza de Oliveira, Maykelin Fuentes Zaldívar, Ana Alice Maia Gonçalves, Lucilene Aparecida Resende, Reysla Maria da Silveira Mariano, Diogo Fonseca Soares Pereira, Ingrid dos Santos Soares Conrado, Mariana Amália Figueiredo Costa, Daniel Ferreira Lair, Diego Fernandes Vilas-Boas, Eiji Nakasone Nakasone, Ingrid de Sousa Ameno, Wanessa Moreira Goes, Denise Silveira-Lemos, Alexsandro Sobreira Galdino, Ronaldo Alves Pinto Nagem, Walderez Ornelas Dutra, Rodolfo Cordeiro Giunchetti

**Affiliations:** 1Laboratory of Biology of Cell Interactions, Department of Morphology, Institute of Biological Sciences, Federal University of Minas Gerais, Belo Horizonte 31270-901, MG, Brazil; dianaso@ufmg.br (D.S.d.O.); mfzaldivar2016@ufmg.br (M.F.Z.); lucilenearesende@ufmg.br (L.A.R.); reyslamariano@ufmg.br (R.M.d.S.M.); diogofsp@ufmg.br (D.F.S.P.); ingrid1709@ufmg.br (I.d.S.S.C.); daniellair@ufmg.br (D.F.L.); diegofervboas@ufmg.br (D.F.V.-B.); eijinakasone@ufmg.br (E.N.N.); ingridameno@ufmg.br (I.d.S.A.); wanessamg@ufmg.br (W.M.G.); dlemos@icb.ufmg.br (D.S.-L.); waldutra@icb.ufmg.br (W.O.D.); 2Laboratory of Biotechnology of Microorganisms, Federal University of São João Del-Rei, Divinópolis 35501-296, MG, Brazil; anaalice-biocel@ufsj.br (A.A.M.G.); asgaldino@ufsj.edu.br (A.S.G.); 3Structural Biology Laboratory, Department of Morphology, Institute of Biological Sciences, Federal University of Minas Gerais, Belo Horizonte 31270-901, MG, Brazil; mariana.amalia@ufmg.br (M.A.F.C.); nagem@icb.ufmg.br (R.A.P.N.)

**Keywords:** visceral leishmaniasis, vaccine, chimeric proteins, patents

## Abstract

The development of prophylactic vaccines is important in preventing and controlling diseases such as visceral leishmaniasis (VL), in addition to being an economic measure for public health. Despite the efforts to develop a vaccine against human VL caused by *Leishmania infantum*, none is available, and the focus has shifted to developing vaccines against canine visceral leishmaniasis (CVL). Currently, commercially available vaccines are targeted at CVL but are not effective. Different strategies have been applied in developing and improving vaccines, such as using chimeric proteins to expand vaccine coverage. The search for patents can be a way of tracking vaccines that have the potential to be marketed. In this context, the present work presents a summary of immunological aspects relevant to VL vaccine development with a focus on the composition of chimeric protein vaccines for CVL deposited in patent banks as an important approach for biotechnological development. The resulting data could facilitate the screening and selection of antigens to compose vaccine candidates with high performance against VL.

## 1. Introduction

### 1.1. General Aspects of Visceral Leishmaniasis

Visceral leishmaniasis (VL) is a neglected disease caused by a protozoan of the *Leishmania donovani* complex (*L. donovani*, *L. infantum* or *L. chagasi*) transmitted by the bite of female sandflies [1]. Due to the level of molecular similarity, *L. chagasi* and *L. infantum* are considered synonymous by some researchers, although other researchers prefer to separate them at a subspecific level, with the names *L.* (*L.*) *infantum infantum* and *L.* (*L.*) *infantum chagasi* [2,3]. VL has a worldwide distribution and has had a major impact on public health in developing countries for decades [4,5], with a predominance of annually occurring new cases in the regions of East Africa, India, and Brazil, and a global incidence of around 50,000 to 90,000 [6]. In 2020, more than 90% of the number of new cases reported were concentrated in 10 countries: Brazil, China, Ethiopia, Eritrea, India, Kenya, Somalia, South Sudan, Sudan, and Yemen [6].

VL caused by *L. infantum* is a zoonotic disease [2], whereupon dogs (*Canis lupus familiaris*) play a prominent role in the parasite’s transmission cycle and maintenance of the parasite by presenting intensive cutaneous parasitism [7,8,9]. Therefore, the proximity of urban *L. infantum*-infected dogs in home environments results in the interaction of the vector with parasitized dogs, demonstrating an increase in the number of VL cases [10].

Euthanasia of seropositive dogs is one of the control methods recommended by Brazil’s Ministry of Health [9], but is considered a controversial measure. There is no consensus on reducing the number of cases in humans and dogs [11,12,13,14,15]. However, the impacts of dog euthanasia and reducing human VL have recently been demonstrated [16]. Early diagnosis and treatment of human cases and control of the insects via the use of insecticides are other recommended measures [9]. Although other countries, such as China, have previously adopted these control measures, Brazil is currently the only country to continue using euthanasia [17,18].

However, these control measures have limitations and present a challenge to reducing the number of canine and human cases [19,20,21]. In this context, the search for the development of effective vaccines against leishmaniasis becomes essential to improving disease control measures worldwide [22,23].

### 1.2. Development of Vaccines against Canine Visceral Leishmaniasis

Despite the existence of candidate vaccines being tested for the development of a vaccine against human VL, they are still not available for use. Additionally, efforts have been made to find more effective vaccines against canine visceral leishmaniasis (CVL) due to the role dogs play in maintaining the parasite transmission [24,25].

Several studies have reported the potential of different vaccine candidates to trigger immunoprotective mechanisms against CVL. Some of these trials are in progress to characterize the protective immunity of candidates for use in a canine vaccine against *L. infantum* infection, including live or dead *Leishmania* parasites, purified *Leishmania* antigens, live recombinant bacteria expressing *Leishmania* antigens, and antigen-encoding DNA plasmid [26,27,28,29,30]. Recently, sandfly antigens have been considered as an important approach to blocking the transmission of VL from infected dogs [23,31,32,33].

The host’s immune response to the parasite may present a pattern of resistance or susceptibility, and the intrinsic characteristics of these profiles are the key to the successful search for vaccines. In the resistance profile, a Th1-type cellular immune response prevails with the production of pro-inflammatory cytokines, such as interferon-gamma (IFN-γ), interleukin-12 (IL-12), tumor necrosis factor-alpha (TNF-α), interleukin-2 (IL-2), and granulocyte-macrophage colony-stimulating factor (GM-CSF) [31,34,35,36,37,38,39]. Nitric oxide (NO) produced by macrophages activated by IFN-γ or TNF-α is a harmful agent to *Leishmania* parasites [40]. In contrast, the predominance of Th2-type cellular immune responses, with the production of cytokines such as IL-4 (interleukin-4), IL-10 (interleukin-10), and transforming growth factor-beta (TGF-β), negatively regulates the activation of phagocytic cells, favoring the proliferation of parasites and disease development [31,41]. The study of susceptibility and resistance biomarkers in CVL could guide the search for a vaccine capable of inducing a protective response. According to this logic, an ideal vaccine against CVL should have immunological features that include the elicitation of a long-lasting cell-mediated immune response that contributes to controlling the infection [21,23,31,42].

#### 1.2.1. Commercially Available Vaccines for Canine Visceral Leishmaniasis

Dogs are the target of studies for the development of vaccines against VL since the control of infection by *Leishmania infantum* is essential to curtailing its spread.

Two vaccines have been made commercially available in Europe: CaniLeish^®^ and LetiFend^®^. CaniLeish^®^ (Virbac S. A., Carros, France) was composed of purified excreted–secreted proteins from *L. infantum* (LiESP) and a purified fraction of saponin *Quilaja saponaria* (QA-21) as an adjuvant. The vaccine protocol consisted of three doses at 21-day intervals and an annual booster [43]. This vaccine induced a Th1-type cellular response with IFN-γ production, a humoral response with increased IgG2, and inducible nitric oxide (iNOS) and nitrogen dioxide synthesis, essential for parasite elimination. A field study showed its effectiveness in preventing clinical signs, reporting that it offered 68.4% and 92.7% vaccine protection [44]. However, CaniLeish^®^ has been off the market since 2021 [45,46]. The LetiFend^®^ (Laboratorios LETI, Barcelona, Spain) vaccine is a more recent vaccine, and unlike the others, it is the only commercial vaccine composed of a chimeric protein containing five antigenic fragments of four *L. infantum* proteins (histone H2A and ribosomal proteins LiP2a, LiP2b, and LiP0), with 72% efficiency [44].

Leish-Tec^®^ (CEVA Saúde Animal, Juatuba, Brazil) was another vaccine option: made commercially available in Brazil in 2015, it was composed of recombinant protein A2 from *L. donovani* amastigote, and contained saponin as an adjuvant [44,47]. The initial vaccination protocol consisted of three doses administered subcutaneously with a 21-day interval between each dose and an annual booster [14]. The A2 recombinant protein found in the amastigote stages of *L. donovani*, *L. amazonensis*, and *L. infantum* induced partial protection against these species in immunized mice [48,49,50]. The vaccine also induced a highly specific humoral immune response in these animals and the cellular immune response was of the mixed Th1/Th2 type [51].

In a canine model study, beagle dogs immunized (*n* = 21) with Leish-Tec^®^ showed high levels of anti-A2 IgG2 antibodies after vaccination, but the vaccine induced only partial protection against *L. chagasi* (syn. *L. infantum*). Clinical signs in immunized and infected dogs appeared late, about one year after infection, as compared to the control group [44,52]. The first field study was conducted in a heterogeneous native canine population (of more than 500 dogs) distributed between vaccine and control groups, with a significant reduction in CVL cases observed in the vaccinated group. However, this study was unable to demonstrate a significant reduction in infectivity in the immunized dogs due to the absence of statistically significant differences in the prevalence of positive sandfly pools after feeding in each experimental group [53]. In another more recent field study, different canine populations were used in control (beagle or mongrel dogs recruited from VL-free areas) and vaccine (dogs from endemic areas) groups. These results demonstrated that the incidence rates of infection among the vaccinated animals (27%, 40/151) and control animals (42%, 33/78) were statistically significant. However, the presence of sick animals in the group of immunized seropositive dogs (44%, 18/40) was double that of the control group (21.2%, 7/33). Although studies have shown that this vaccine induces a significant increase in IFN-γ levels, as well as in the humoral immune response with IgG2 antibodies, and a partial protective response against the parasite, it lacks evidence of its efficacy [14,44,54]. Leish-Tec^®^ was suspended in 2023 after it was found that its A2 protein content was lower than the minimum limit established in the Product License [55].

Despite several studies on licensed vaccines for CVL, methodological limitations, such as the lack of standardization in experimental design, can make it difficult to compare the effectiveness of these vaccines [44]. Therefore, there remains a pressing need for research to test vaccines and develop new vaccine candidates against CVL.

#### 1.2.2. Chimeric Proteins Used as Potential Vaccine Candidates for Visceral Leishmaniasis

Due to the variability of the parasite and its interaction with a mammalian immune system, characterized by its genetic polymorphism, some studies have suggested that vaccines composed of polypeptides would be more likely to induce protective immune responses against leishmaniasis in different individuals [56,57]. In this context, the use of vaccines consisting of recombinant proteins would have a low cost, standardized achievement, and stability [58].

“Chimaera” is a Greek word used to refer to a creature with the head of a lion, the body of a goat, and the tail of a snake. Inspired by this mythological figure, chimeric proteins are those whose construction is based on two or more individual proteins or peptides fused together, producing a single polypeptide chain [59,60].

The first candidate for a chimeric protein vaccine against VL was KSAC, a polyprotein comprising kinetoplastid membrane protein 11 (KMP11), sterol 24-c-methyltransferase (SMT), amastigote protein A2, and cysteine proteinase B (CPB). A study that evaluated this vaccine found that it was able to induce antigen-specific multifunctional Th1-type cells, protecting the mice used in the study against the *L. infantum* challenge. The induction of pro-inflammatory cytokines, such as TNF-α and IFN-γ, has also been observed [56]. Furthermore, this vaccine significantly reduced the parasitic burden on the animals’ spleen and liver. The choice of these proteins was based on their known protective efficacy against VL [61,62,63,64]. LetiFend^®^ (Laboratorios LETI, Barcelona, Spain) is the only commercially available vaccine composed of a chimeric protein [44].

Although the protective action of these proteins is recognized, it is not guaranteed to trigger efficient protection, either in the human or canine population, due to the difference between those host immune responses [56]. Therefore, numerous strategies have been used to identify antigens to compose and improve vaccines, such as the use of bioinformatics tools (reverse vaccinology). These tools provide a diversification of vaccines, and those that include polyepitopes are being explored [65]. In this context, the present review aimed to identify patents for prophylactic vaccines consisting of chimeric proteins as potential candidates against VL.

## 2. Patents of Chimeric Proteins (2010–2023)

Patent documents protect and grant intellectual property rights that contain important commercial, technological, and scientific information in technological development [66,67]. Also, unlike peer-reviewed scientific articles, patents provide important information on possible short-term marketable products and their more direct applicability [67].

A patent search was conducted using the tools from the National Institute of Industrial Property (INPI) [68], Espacenet [69], and the World Intellectual Property Organization (WIPO) [70] databases. The keywords were Leishmani*, vaccine, chimer*, multi or polyepitope, truncated, combined with Boolean operators “AND” and “OR”, and/or the codes of the International Patent Classification (ICP). Truncation allows the localization of words with a common stem but different suffixes, represented by the character (*). The codes were selected according to the WIPO classification, which allowed the association of patents linked to vaccines against VL. The codes used were subgroup A61K 39/008 (“Medical preparations containing antigens or antibodies—*Leishmania* antigens”), A61P 33/02 (“Antiparasitic agents—antiprotozoals”), C07K 14/44 (“Peptides having more than 20 amino acids—protozoa”), G01N 33/569 (“Investigating or analyzing materials for microorganisms, e.g., protozoa, bacteria, viruses”), and C12N 15/30 (“Mutation or genetic engineering; DNA or RNA concerning genetic engineering; and vectors, e.g., plasmids, or their isolation, preparation or purification—genes encoding protozoal proteins, e.g., from *Plasmodium*, *Trypanosoma*, *Eimeria*”).

Selected patents contained the results of prophylactic vaccines composed of chimeric proteins and included in vivo and/or in vitro assays.

Nine patents made up of chimeric proteins were found from 2010 to 2022: four from Brazil, three from the United States of America, and two from Spain. All of the patents included here were developed by bioinformatics tools for identifying proteins that presented epitopes for human and/or mice T cells as a prerequisite to the construction of chimeric proteins. The results demonstrated that they have a predominantly Th1-type profile, with the induction of proinflammatory cytokines and reduced parasitic load (when evaluated).

The patents are summarized in Table 1, including the country of priority filing, the chimeric vaccine proteins, and the summary of results.

### Protein Targets

The chimeras described in Table 1 primarily consist of known proteins with recognition of the major histocompatibility complex (MHC), in addition to previous studies demonstrating their potential to be used in vaccines against leishmaniasis, either individually or in association with other proteins. The proteins and some findings of their protective potential against leishmaniasis have been described below and their localization is shown in the schematic picture (Figure 1).

Cysteine proteases (CPs) are enzymes with important functions in the pathogenesis of several parasites during their interaction with the host. They are associated with cell and tissue invasion, protein hydrolysis, autophagy, and modulation of the host’s immune response [80]. The CPA, CPB, and CPC subtypes have been explored as vaccine candidates, capable of inducing protective immunity against cutaneous and visceral leishmaniasis, with the induction of NO and IFN-γ production in mice [81,82,83]. In addition, CPs were able to induce the production of IFN-γ, TNF-α, and IgG2 and low production of IgG1 antibodies and IL-10 in vaccinated dogs [84].

The heat shock protein HSP70 is a ubiquitous 70kDa molecular chaperone, highly conserved, and important in the folding and remodeling processes of cellular proteins [85]. It can thus be present in mitochondria and the endoplasmic reticulum [86]. Its expression is high during the process of passing from the invertebrate vector to the mammalian host, which is important in differentiating between the forms of the parasite [87]. HSP70 can induce high levels of IgG2a, IFN-γ, and IL-2 in mice immunized against *L. donovani* and suppress the Th2-type immune response [88], as well as the maturation capacity of splenic dendritic cells in mice [89].

K39 is an immunodominant epitope in kinesin-related proteins, consisting of 39 amino acids, highly conserved in the *Leishmania donovani* complex, present in amastigote forms, and involved in the intracellular process [90,91]. This protein has been used to detect anti-*Leishmania* antibodies in several diagnostic platforms with a high degree of accuracy. The Kalazar Detect™ (InBios International, Seattle, WA, USA), the IT LEISH^®^ (BIO-RAD Laboratories Inc., Marnes-la-Coquette, France), and the OnSite™ *Leishmania* IgG/IgM Combo test (CTK Biotech, Poway, CA, USA) are immunochromatographic tests that have been used in the diagnosis of human VL and are available in the Brazilian public health system [92]. Furthermore, the Dual-Path Platform (DPP—Bio-Manguinhos/Fiocruz, Rio de Janeiro, Brazil), presents a fused protein of rK26/rK39 used for the diagnosis of CVL, also available in Brazil [93]. Regarding immunoenzymatic assays, there are several studies evaluating the performance of rK39 in the diagnosis of both human and canine VL [94,95,96,97]. Despite its wide use in the diagnostic field, only one patent (BR102021000794) [71] was found that used rK39 as a possible vaccine candidate.

Kinetoplastid membrane protein 11 (KMP11) is mainly expressed on the cell surface of amastigotes and promastigotes, but it is also found in membrane structures, intracellular vesicles, and flagellar pockets [98]. This protein can produce strong antigenicity for T cells in humans and mice, being a candidate for the vaccine [99]. Its expression is increased in metacyclogenesis and with greater expression in the amastigote form [100]. Results of studies with this protein have revealed its ability to produce IFN-γ, IL-10, and IgG1 and IgG2a antibodies. Despite stimulating the production of IL-10 and IgG1, their levels were lower than those of the IgG2a and IFN-γ, indicating a more prominent activation of the Th1-type response, resulting in a parasite load reduction in the spleen and lymph nodes in mice [101].

*Leishmania* homolog of activated C kinase (LACK) is a highly conserved protein, an immunodominant present in all *Leishmania* species, and expressed in both amastigote and promastigote forms [102,103]. Studies have demonstrated its ability to induce CD8+ T cells and IFN-γ in *L. major*-infected mice [104]. In addition, the production of IFN-γ and IL-10 by peripheral blood mononuclear cells (PBMCs) was reported in patients with cutaneous leishmaniasis caused by *L. amazonensis*, *L. guyanensis,* and *L. braziliensis* [105,106,107,108]. Similarly, asymptomatic individuals and VL-cured patients presented the production of IFN-γ and TNF-α [29,109]. LACK induced the production of IL-4 [110], and different approaches are needed to redirect the initial IL-4 responses to Th1, such as the use of cytokines or DNA vectors [111,112]. However, some authors have reported failure to protect mice against *L. infantum* and *L. donovani*, even when using a DNA plasmid expressing LACK for immunization [113,114]. In contrast, Fernández et al. [115] reported promising results with the use of LACK together with the attenuated Vaccinia virus, in which the protein was recognized by the T cells of asymptomatic individuals infected with *L. infantum* and VL-cured patients. These data on the LACK antigen reported contradictory results and did not support its use in vaccine formulations against VL.

LiP2a and LiP0 are acidic ribosomal proteins of *L. infantum* characterized by being immunodominant antigens recognized by the serum of humans and other animals infected by *L. infantum* [116,117]. These proteins were able to trigger the immune response and induce protection against infection by *L. infantum* and *L. major* in mice, with increased CD4+ and CD8+ T cells, and significant production of antigen-specific IL-12 [118]. These proteins also stimulate IFN-γ production via splenocytes in mice immunized with LiP2a [119].

Hypothetical proteins have been described in the genome of *Leishmania* spp. but without a defined biological role [120]. Through an immunoproteomic study, LiHyp1 and LiHyp2 were recognized by antibodies in serum from VL patients but not in serum of healthy individuals [121]. The use of hypothetical proteins is still a field to be explored. Some authors have demonstrated that hypothetical proteins are capable of reducing the parasitic load on the liver, spleen, bone marrow, and lymph nodes, with an immunogenic profile related to high levels of IFN-γ, IL-12, GM-CSF, and specific IgG2 production [122,123].

Nucleoside hydrolases (NHs) are vital enzymes in the metabolism of DNA, being essential for the replication of parasites, especially during the initial stages of infection. These enzymes are present in all species of *Leishmania* spp., justifying their use as phylogenetic markers. Additionally, they share high identification levels among many microorganisms but are absent in mammals [124]. These characteristics make NHs targets for an anti-VL vaccine by inducing high immunogenicity [124]. For instance, the NH36 of *L. donovani* is a non-specific nucleoside hydrolase that is the main antigen of Leishmune^®^, a vaccine previously sold in Brazil from 2004 to 2014, which was discontinued due to noncompliance with the requirements of phase III studies for efficacy (Brazil, Technical Note 038/2014). This vaccine had an efficacy and protection rate greater than 80%, being able to induce high levels of IgG2, a predominantly Th1-type immune response with high production of IFN-γ, TNF-α, and IL-17 [44]. Other studies have affirmed the potential of NHs [124,125].

Nucleosomal histones are important proteins in the DNA packaging process, transcription, and gene regulation. Evidence suggests that histones from *Leishmania* spp. are relevant immunogens during parasite/host interactions [126]. Therefore, histones H2A, H2B, H3, and H4 have been studied as potential vaccines against leishmaniasis. A profile similar to the Th1 response was detected in murine models immunized with histones against *L. major* and *L. donovani* infections with IFN-γ and TNF-α induction and low IL-4 production [126,127,128,129]. Moreover, this immunization was able to stimulate immune responses in the PBMCs of cured individuals and patients infected with *Leishmania*, as well as reduce the parasitic burden by more than 80% in the spleen, liver, and bone marrow in hamsters [129].

Prohibitins (PHBs) are conserved proteins found in all eukaryotic cells in the inner membrane of the mitochondria. These proteins are important in several functions linked to this organelle and the stabilization of its membrane. In *Leishmania* spp., prohibitins are involved in cell proliferation, greater infectivity, and maintenance of the parasite’s mitochondrial integrity [130,131]. The presence of anti-PHB antibodies in patients infected with *L. donovani* demonstrated that these proteins are relevant when the disease is active [130]. A study by Lage et al. [132] verified the induction of a Th1-type cellular response with high levels of IFN-γ, IL-12, and GM-CSF in immunized animals. Moreover, a significant reduction in the parasite load was reported in the spleen, liver, lymph nodes, and bone marrow in mice. High levels of IFN-γ in PBMC from healthy individuals and cured VL patients have also been observed [133].

Promastigote surface antigens (PSAs) are members of a family of membrane-bound or secreted proteins from *Leishmania* spp., involved in resistance to lysis promoted by the complement system during interactions with host cells [134,135]. PSAs are highly conserved and, despite being recognized by immune response cells, preferably Th1 in humans, they do not share homology with mammalian cells [136]. Studies with PSA subtypes have shown that they are capable of inducing Th1-type responses with IFN-γ production in mice against *L. major* infection [137] and proliferation of PBMCs in patients with cutaneous leishmaniasis in response to PSA-2 [138]. Chamakh-Ayari et al. [136] demonstrated that the PSA-38S of *L. amazonensis* induced a mixed Th1/Th2 response in individuals with immunity to *L. major* and *L. infantum*, in addition to inducing granzyme B production. Petitdidier et al. [139] demonstrated that PSA can induce IFN-γ, nitric oxide, and IgG2 antibodies in vaccinated dogs.

Small glutamine-rich TPR proteins (SGTs) are co-chaperones that interact with HSP70 and HSP90 chaperones to ensure their proper functions, which are essential to the parasite’s life cycle and survival [140]. Dias et al. [141] identified the potential of *L. infantum*’s SGTs for vaccine and diagnostic approaches against VL. Mice immunized with SGTs developed a specific Th1-type response by producing IFN-γ, IL-12, and IgG2a, which induced a resistance profile against the infection.

Specific amastigote protein A2 corresponds to a family of specific amastigote genes necessary for the parasite’s survival and results in a virulence factor [142]. A2 was the main protein in the Leish-Tec^®^ vaccine and has been explored as a possible vaccine candidate as it has been shown to induce a Th1 immune response represented by biomarkers IFN-γ, TNF-α, and IgG2 antibodies that conferred partial protection against CVL [14,47,52,143,144].

Sterol 24-c-methyltransferase (SMT) is an enzyme of the transferase family, important in steroid production, especially ergosterol biosynthesis present in the *Leishmania* spp. membrane [145]. Mice immunized with SMT showed IFN-γ induction and lower spleen and liver parasite loads [64]. SMT, together with the nucleoside hydrolase NH36 of *L. donovani,* composed a vaccine developed against human VL that reached phase I clinical trials [146].

## 3. Virus-like Particles—VLPs

A virus-like particle (VLP) is based on a type of vaccine composed of protein subunits derived from viruses, thus forming a non-pathogenic particle without the viral genome. VLPs are highly immunogenic with a similar immune response capacity to that of a natural viral infection, but in a non-infectious way [147]. Furthermore, VLPs are versatile and have a great biodiversity and can come from single-stranded DNA viruses, positive or negative sense RNA, in varying sizes, with varying complexity in terms of structure, in monolayers or multiple layers, or in the presence of an envelope, among others [147,148,149,150]. As in this present review, some VLPs can form polyvalent or mosaic VLPs, thus allowing them to be composed of multiple viral strains [151] in order to facilitate more complex vaccine constructions [152], modifications or insertions of sequences of proteins or peptides, and the formation of recombinant VLP chimeras containing xenogeneic antigens [147].

Many VLPs may have structures or molecules in their composition with self-stimulating characteristics, triggering an immune response as an adjuvant itself [147,153]. However, the use of adjuvants in the vaccine allows the targeting of a specific type of immune response that is desired [147].

Given these advantages and great plasticity, several VLP-based vaccines are commercially available for diseases caused by viruses such as human papillomavirus (HPV), with Gardasil (Merck, Darmstadt, Germany) and Cervarix (GlaxoSmithKline, Brentford, UK) both being available in Brazil. For hepatitis B (HBV), there are the Indian vaccines Elovac B (Indian Immunologicals LTD, Telangana, India), Genevac B (Serum Institute of India PVT. Ltd., Pune, India), and Shanvac B (Shanta Biotechnics LTD., Hyderabad, India). The German Recombivax HB (Merck, Darmstadt, Germany) and the British Engerix-B (GlaxoSmithKline, Brentford, UK) are also found in Brazil. Hecolin (Xiamen Innovax Biotech Co., Ltd., Xiamen, China) is available against hepatitis E (HEV) [147].

Regarding the leishmaniasis, two studies were developed using VLPs. The first study used the α-Gal trisaccharide coupled to the Qβ VLP. This carbohydrate is found on the surface of *Leishmania* spp. related to the virulence and evasion of the parasite [154]. Using C57BL/6 knockout mice for α-galactosyltransferase to mimic the biochemistry of α-Gal in humans, the researchers observed that after immunization there was protective immunity against *L. infantum* and *L. amazonensis* challenges. Furthermore, the control of the parasite infection in the spleen and liver demonstrated the potential of VLPs as a vaccine candidate against visceral and cutaneous leishmaniasis [154].

The second study included the patent BR1020160254493A2 [155], based on a multivalent vaccine consisting of three recombinant proteins derived from the parasite and the vector: namely KMP11 and LeishF3 from the parasite and LJL143 from saliva of *Lutzomyia longipalpis*, in association with the lipid adjuvant glucopyranosyl (GLA-SE), a TLR4 agonist [156]. LJL143 is one of the sandfly salivary proteins studied against VL in dogs [157]. LeishF3 is a fusion protein composed of nucleoside hydrolase, sterol 24-c-methyltransferase, and cysteine protease B. In this study, the authors reported an increase in the CD4+ and CD8+ T cell proliferation in an ex vivo assay using vector and parasite antigens [156].

## 4. Discussion and Future Perspectives

Vaccines are considered essential tools in preventing VL. Classic vaccines against leishmaniasis are based on live or attenuated parasites or their subunits. Additionally, the search for more specific vaccines could be more promising using *Leishmania* peptides that display high immunogenicity and protection profiles. As discussed here, the development of vaccines consisting of chimeric proteins could be more effective by linking several of the parasite’s antigens in a peptide chain. The genetic polymorphism of mammals and the variability of parasites would no longer be a hindrance. In addition, peptides offer advantages such as good stability, the absence of potentially harmful materials, low antigen complexity, and low cost of amplification.

The construction of these proteins is only possible with the use of reverse vaccinology using bioinformatics tools. This approach can predict specific T cell epitopes, an important action for targeting VL antigens and triggering an effective adaptive immune response. This is why chimeric vaccines show so much promise [158]. However, peptides are poorly immunogenic and the use of a potent adjuvant or nanotechnology via antigen encapsulation could overcome this drawback.

Chimeric vaccines must be constructed from *Leishmania* spp. associated with parasite invasion and survival in the vertebrate host (mammals) [159]. As has been demonstrated, the vaccines discussed here have different compositions and, with the exception of patent number. BR1020160061210 [75], composed of hypothetical proteins (unknown), all contain one or more proteins related to the invasion, survival, and/or metabolic processes important for parasites in the vertebrate host.

Regarding the results described here, all vaccines composed of chimeric proteins induce high levels of proinflammatory cytokines, mainly IFN-γ, in cell cultures, while reducing the parasitic load of the spleen, liver, and/or lymph nodes. Furthermore, the compositions of patent BR1020180081977 [73] were able to induce CD4+ and T CD8+ T cells with a central memory phenotype.

The patents described in this study could guide the choice of proteins that have great potential for use as a promising vaccine to control VL. Alternatively, the VLPs could provide an important vaccine formulation able to overcome limitations such as the selection of the suitable adjuvant able to trigger a protective immune response against VL. The studies described reinforce the need for additional vaccine formulations using different targets, including parasite and sandfly antigens capable of (i) interfering with the life cycle of the insect vector and (ii) triggering a protective immune response against the parasite in dogs, resulting in effectively blocking the transmission of *L. infantum*.

## 5. Conclusions

The patents reporting the use of *Leishmania* antigens demonstrated that vaccines composed of chimeric proteins have the potential to control VL. Therefore, this review is offered as a way to facilitate the identification and selection of chimeric proteins for the improvement and development of new VL vaccines.

## Figures and Tables

**Figure 1 vaccines-12-00271-f001:**
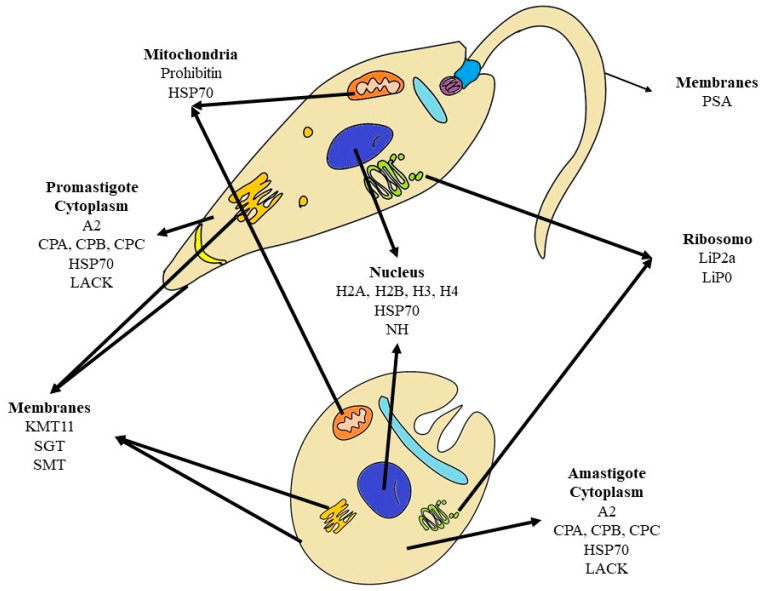
Schematic picture of promastigote and amastigote forms of *Leishmania* spp. and location of proteins described in patents related to vaccines composed of chimeric proteins against visceral leishmaniasis.

**Table 1 vaccines-12-00271-t001:** Identification of patents related to vaccines composed of chimeric proteins for VL.

Original Patent Title	Publication Number	Priority Country	Deposit Year	Chimera Composition	In Silico Analysis	In Vitro Analysis	In Vivo Analysis
“Proteína quimérica, kit, método para diagnóstico de leishmaniose, uso de proteína quimérica, composição vacinal contra leishmaniose visceral, e, uso de uma composição vacinal”	BR 10 2021 00079 4 [71]	Brazil	2021	A2 and K39	-	↑IFN-γ in spleen cell culture	↑IgG e IgG2, in BALB/c mice. ↓spleen and liver, parasite load by limiting dilution, in BALB/c mice
“Quimera sintética multiepitópica como vacuna y tratamiento frente a leishmaniosis en mamíferos”	ES2795149 [72]	Spain	2020	H2A, H2B, H3, and H4	Human and mice MHC * class I and II alleles prediction	↑IFN-γ and IL-12 after culture stimulation. ↑leishmanicidal effect in infected BMDC **	↓spleen and liver, parasite load by limiting dilution, in BALB/c mice
“Vacinas compostas de proteínas quiméricas poliepítopos contra a leishmaniose visceral humana e/ou canina”	BR 10 2018 008197 7 [73]	Brazil	2018	VAC-1: H2A, LACK LiP2a, LiP0, and CPC	MHC class I and II alleles prediction	↑IFN-γ, TNF-α, CD4+ T cells, and CD8+ T cells, after culture stimulation ↑CD4+ T lymphocytes with central memory phenotype in VAC-1 and VAC-2 ↑CD8+ T lymphocytes with central memory phenotype in VAC-1 and VAC-2. Effector memory phenotype in VAC-1	↓spleen parasite load by Real Time PCR (qPCR), in BALB/c mice
				VAC-2: CPA, CPB, PSA-50S, and A2			
“Proteína quimérica recombinante, vacina contra leishmanioses e uso”	BR 10 2017 025621 9 [74]	Brazil	2017	Prohibitin; SGT; LiHyp5	Human MHC class I and II alleles prediction	↑IFN-γ, IL-12, and GM-CSF in spleen cell culture ↑PBMCs proliferation in human and dog cells	↓spleen, liver, draining lymph nodes, and bone marrow parasite load by limiting dilution, in BALB/c mice
“Proteína quimérica, composição vacinal contra leishmanioses e usos”	BR 10 2016 006121 0 [75]	Brazil	2016	LiHyp1, LiHyp6, LiHyV, and HRF	Human and mice MHC class I and II alleles prediction	↑IFN-γ, IL-12, and GM-CSF in spleen cell culture	↓spleen, liver, draining lymph nodes, and bone marrow parasite load by limiting dilution, in BALB/c mice
Recombinant polyprotein vaccines for the treatment and diagnosis of leishmaniasis	US20130177584 [76]	United States	2013	KSA (KM11, SMT e A2)	-	-	↓liver by limiting dilution, in C57BL/6 (*L. infantum* challenge), and BALB/c mice (*L. donovani* challenge)
Vaccines comprising leishmania polypeptides for the treatment and diagnosis of leishmaniasis	WO 2014/160987 [77]	United States	2013	NS and NSC	-	↑IFN-γ in spleen cell culture	↓liver by qPCR, in BALB/c mice
“Quimera multicomponente para su uso como vacuna frente a la infección por *Leishmania* spp. En mamíferos”	WO 2013/110824 [78]	Spain	2011	HISA70 (H2A, H2B, H3, H4, A2, HSP70)	-	-	↓spleen and liver by limiting dilution, in BALB/c mice
Vaccines comprising non-specific nucleoside hydrolase and sterol 24-c-methyltransferase (SMT) polypeptides for the treatment and diagnosis of Leishmaniasis	WO 2012064659 [79]	United States	2010	NS	-	↑IFN-γ by spleen cell culture, in BALB/c mice ↑IFN-γ, and ↑IgG in non-human primates	↓liver by limiting dilution, in BALB/c mice ↑IgG1 and IgG2 titration

The arrows (↓ and ↑) indicate the decrease and increase in biomarker levels and/or parasite load. * Major histocompatibility complex (MHC). ** Bone marrow-derived dendritic cells (BMDCs). A2: specific amastigote protein; CPA, CPB, and CPC: cysteine peptidase proteins; HRF: IgE-dependent histamine-releasing factor; HSP70: heat shock protein; H2A, H2B, H3, H4: nucleosomal histones; K39: kinesin-related protein; KSA and KSAC: kinetoplastid membrane protein (KM11), sterol 24-c-methyltransferase (SMT), specific amastigote protein A2, and cysteine peptidase proteins B; LACK: activated kinase C receptor homologous *Leishmania* protein; LiHyp 1, LiHyp5, LiHyp6, and LiHyV: hypothetical proteins; LiP2a and LiP0: acidic ribosomal proteins; NS: non-specific nucleoside hydrolase (NH) and sterol 24-c-methyltransferase (SMT); NSC: non-specific nucleoside hydrolase (NH), sterol 24-c-methyltransferase (SMT), and cysteine polypeptidase B (CPB); prohibitin: surface protein; PSA-50S: promastigote surface antigen; SGTs: small glutamine-rich TPR proteins.

## Data Availability

All collected data are reported in the text.

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
