# Peer review of "New Approaches to the Prevention of Visceral Leishmaniasis: A Review of Recent Patents of Potential Candidates for a Chimeric Protein Vaccine"

_vaccines, 2024, doi:10.3390/vaccines12030271_

Round 1
Reviewer 1 Report
Comments and Suggestions for Authors
This is an interesting and nicely written review in general and with an extensive, useful bibliography. The authors identify the problems of Leishmania vaccines and then subsequent areas that may be fruitful to develop such as use of chimeric protein to develop the vaccine. They also focus on developing canine vaccines as an ‘easier’ route to helping control parasite transmission. Table 2 is useful for reviewing patents in this area. This discussions on the protein targets for vaccines is also very useful. The authors should, however, consider some grammar/sentence structure modifications as indicated:
1. Line 20 implies this is already done rather than there is an important need for vaccine development
2. Line 43: add ‘annually following new cases
3. Line 49: the meaning is unclear
4. Line 107: remove ‘has been’ and add ‘…is composed….’
5. Line 129 is awkward and thus revise
6. Line 135: what does ‘they’ refer to?
Comments on the Quality of English Language
only a few suggestions about the language quality
Author Response
We are thankful to Reviewer #1 for critical reading and for contributing to improving the quality of the manuscript. The manuscript was revised according to the Reviewer #1 suggestions and the changed sentences appears as follows:
- Line 20: Prophylactic vaccines development are important in preventing and controlling diseases such as visceral leishmaniasis (VL), in addition to being economic measures for public health.;
- Line 43 (current lines 41-44): we accepted the suggestion: “(…) VL has a worldwide distribution and has represented a major impact on public health in developing countries for decades [4, 5] with a predominance of ‘annually following new cases in the regions of East Africa, India, and Brazil, and a global incidence of around 50,000 to 90,000 [6]. (…)”
- Line 49 (current lines 49-51): “(…) Therefore, the proximity of urban L . infantum-infected dogs in home environments results in the interaction of the vector with parasitized dogs, demonstrating an increase in the number of VL cases [10]. (…)”.
- Line 107 (current line 108): “(…) Leish-Tec® (CEVA, Brazil), is composed commercially available in Brazil since 2015, (…)”.
- Line 129 (current lines 129-130): “(…) However, the presence of sick animals in the group of immunized seropositive dogs (44%, 18/40) was double that of the control group (21.2%, 7/33). (…)”.
- Line 135 (current lines 136-138): “(…) Despite several studies on licensed vaccines for CVL, methodological limitations, such as the lack of standardization in experimental design, can make it difficult to compare the effectiveness of these vaccines. (…)”.
Reviewer 2 Report
Comments and Suggestions for Authors The review paper by Dr Giunchetti's group on vaccines against Visceral Leishmaniasis is good in summarizing the state-of-art the field, especially on the immunological aspects relevant to VL vaccine development. The main focus of this review is the composition of chimeric protein vaccines for canine visceral leishmaniasis. The overall layout and writing style are well organized. The reviewer has some minor suggestions for autors to improve their paper. 1) Where is current status of the VLP vaccines for visceral leishmaniasis in canine and other animals? Why this important section is not included in this review paper. The authors should have some explainations in the beginning of the paper. 2) Increase the size of arrowheads used in Fig. 1. This reviewer would also encourage the authors to include the possible functions of those subunit vaccine candidates. 3) The Discussion and Future perspectives are not well balanced in this review. This reviewer (also as readers) would appreciate if the authors could provide more opinions from their own. For example, the authors should stated in detail that 1) how to seleted subunit/VLP vaccines with/without adjuvants; 2) the proper combination of subunit vaccines and design various VLP with multiepitops displayed; 3) How to solve the low effcacy in transformation from LAB to industry market. 4) The English and writing style need proofreading before resubmission. Comments on the Quality of English LanguageSome changes and proofreading are needed before resubmission.
Author Response
We thank Reviewer #2 for critical reading and for contributing to the quality of the manuscript with such relevant information about VLPs. Considering your observations and the importance of VLPs in vaccine development, we have included a section for providing an overview of the use of VLPs containing the vaccines that used this technology and are commercially available. Furthermore, we mention the development of a vaccine using VLP for visceral leishmaniasis, focusing mainly on its use in humans. The revised manuscript including the new section/paragraph are described below:
“(…) 3. Virus-Like Particles - VLPs
Virus-Like Particle (VLP) is based on a type of vaccine composed of protein subunits derived from viruses, thus forming a non-pathogenic particle, without the viral genome. VLPs are highly immunogenic with a similar immune response capacity to that of a natural viral infection, but in a non-infectious way [138]. Furthermore, VLPs are versatile and have a great biodiversity and can come from single-stranded DNA viruses, positive or negative sense RNA, varying sizes, complexity in terms of structure, whether monolayers or multiple layers, presence of an envelope, among others [138–141]. As in this present review, some VLPs can form polyvalent or mosaic VLPs, thus allowing them to be composed of multiple viral strains [142] in order to facilitate more complex vaccine con-structions [143], modifications or insertions of sequences proteins or peptides and the formation of recombinant VLP chimeras containing xenogeneic antigens [138].
Many VLPs may have structures or molecules in their composition with self-stimulating characteristics, triggering an immune response as an adjuvant itself [138, 144]. However, the use of adjuvants in the vaccine allows targeting a specific type of immune response that is desired [138].
Given these advantages and great plasticity, several VLP-based vaccines are com-mercially available for diseases caused by viruses such as human papillomavirus (HPV), with Gardasil (Merck, Germany) and Cervarix (GlaxoSmithKline, England), both available in Brazil. For hepatitis B (HBV) there are the Indian vaccines Elovac B (Human Biologicals Institute, India), Genevac B (Serum Institute, India) and Shanvac B (Shanta Biotechnics). The German Recombivax HB (Merck), and the British Engerix-B (Glax-oSmithKline) are also found in Brazil. Hecolin (Innovax, China) is available against hepatitis E (HEV) [138].
Regarding the leishmaniasis, two studies were developed using VLPs. The first study used the α-Gal trisaccharide coupled to the Qβ VLP. This carbohydrate is found on the surface of Leishmania spp related to the virulence and evasion of the parasite [145]. Using C57BL/6 knockout mice for α-galactosyltransferase, to mimic the biochemistry of α-Gal in humans, the researchers observed that after immunization there was protective immunity against L. infantum and L. amazonensis challenges. Furthermore, the control of the parasite infection in the spleen and liver demonstrated the potential the VLPs as vaccine candidate against visceral and cutaneous leishmaniasis [145].
The second study included a patent BR1020160254493A2, based on a multivalent vaccine consisting of three recombinant proteins derived from the parasite and the vector, namely: KMP11 and LeishF3 from the parasite and LJL143 from saliva of Lutzomyia longipalpis, in association with the lipid adjuvant glucopyranosyl (GLA-SE), a TLR4 agonist [146]. The LJL143 is one of the sandfly salivary proteins studied against VL in dogs [147]. LeishF3 is a fusion protein composed of nucleoside hydrolase, sterol 24-c-methyltransferase and cysteine protease B. This study the authors reported an in-crease in the CD4+ and CD8+ T cells proliferation in the ex vivo assay using vector and parasite antigens [146]. (…)”.
In the discussion section, the following paragraph was included:
“(…) The patents described in this study could guide the choice of the proteins that have great potential as promising vaccine to control VL. Alternatively, the VLPs could provide an important vaccine formulation able to overcome limitations such as the selection of the suitable adjuvant able to trigger a protective immune response against VL. The studies described reinforces the need for additional vaccine formulations using different targets, including parasite and sandfly antigens capable of (i) interfering in the life cycle of the insect vector and (ii) triggering a protective immune response against the parasite in the dog, resulting in effectively blocking the transmission of L. infantum. (…)”.